# Combining antibiotics with antivirulence compounds can have synergistic effects and reverse selection for antibiotic resistance in *Pseudomonas aeruginosa*

Chiara Rezzoagli[1,2☯], Martina Archetti[1,2☯], Ingrid Mignot[1], Michael Baumgartner[3], Rolf Kümmerli[1,2]*

1 Department of Quantitative Biomedicine, University of Zurich, Zurich, Switzerland, 2 Department of Plant and Microbial Biology, University of Zurich, Zurich, Switzerland, 3 Institute for Integrative Biology, Department of Environmental Systems Science, ETH Zurich, Zurich, Switzerland

☯ These authors contributed equally to this work.
* rolf.kuemmerli@uzh.ch

**Data Availability Statement:** All data are available from the Figshare depository (https://doi.org/10.6084/m9.figshare.12515364).

## Abstract

Antibiotics are losing efficacy due to the rapid evolution and spread of resistance. Treatments targeting bacterial virulence factors have been considered as alternatives because they target virulence instead of pathogen viability, and should therefore exert weaker selection for resistance than conventional antibiotics. However, antivirulence treatments rarely clear infections, which compromises their clinical applications. Here, we explore the potential of combining antivirulence drugs with antibiotics against the opportunistic human pathogen *Pseudomonas aeruginosa*. We combined two antivirulence compounds (gallium, a siderophore quencher, and furanone C-30, a quorum sensing [QS] inhibitor) together with four clinically relevant antibiotics (ciprofloxacin, colistin, meropenem, tobramycin) in 9×9 drug concentration matrices. We found that drug-interaction patterns were concentration dependent, with promising levels of synergies occurring at intermediate drug concentrations for certain drug pairs. We then tested whether antivirulence compounds are potent adjuvants, especially when treating antibiotic resistant (AtbR) clones. We found that the addition of antivirulence compounds to antibiotics could restore growth inhibition for most AtbR clones, and even abrogate or reverse selection for resistance in five drug combination cases. Molecular analyses suggest that selection against resistant clones occurs when resistance mechanisms involve restoration of protein synthesis, but not when efflux pumps are up-regulated. Altogether, our work provides a first systematic analysis of antivirulence-antibiotic combinatorial treatments and suggests that such combinations have the potential to be both effective in treating infections and in limiting the spread of antibiotic resistance.

## Introduction

Scientists together with the World Health Organization (WHO) forecast that the rapid evolution and spread of antibiotic resistant bacteria will lead to a worldwide medical crisis [1–3].

**Funding:** This project has received funding from the Swiss National Science Foundation (grant no. 31003A_182499 to RK) (http://www.snf.ch/en/Pages/default.aspx) and the European Research Council (ERC) under the European Union's Horizon 2020 research and innovation programme (grant agreement no. 681295 to RK) (https://erc.europa.eu). The funders had no role in study design, data collection and analysis, decision to publish, or preparation of the manuscript.

**Competing interests:** The authors have declared that no competing interests exist.

**Abbreviations:** AHL, N-acylhomoserine lactone; AtbR clones, antibiotic resistant clones; CAA, casamino acid medium; CAS, casein medium; ENA, European Nucleotide Archive; ESKAPE, group of pathogens including *Enterococcus faecium*, *Staphylococcus aureus*, *Klebsiella pneumoniae*, *Acinetobacter baumannii*, *Pseudomonas aeruginosa*, *Enterobacter* spp; GFP, green fluorescent protein; IC$_{50}$, half maximal inhibitory concentration; INDEL, insertion or deletion; LB, Lysogeny broth; MIC, minimal inhibitory concentration; OD$_{600}$, optical density at 600 nm; PBS, phosphate buffer saline; PI, propidium iodide; QS, quorum sensing; SNP, single nucleotide polymorphism; Tf, human apo-transferrin; WHO, World Health Organization; WT, wild-type.

Already today, the effective treatment of an increasing number of infectious diseases has become difficult in many cases [4,5]. To avert the crisis, novel innovative approaches that are both effective against pathogens and robust to the emergence and spread of resistance are urgently needed [6,7]. One such approach involves the use of compounds that disarm rather than kill bacteria. These so-called "antivirulence" treatments should exert weaker selection for resistance compared with classical antibiotics because they simply disable virulence factors but are not supposed to affect pathogen viability [8–10]. However, a downside of antivirulence approaches is that the infection will not necessarily be cleared. This could be particularly problematic for immuno-compromised patients (patients with AIDS, cancer, cystic fibrosis, and intensive-care unit patients), whose immune systems are conceivably too weak to clear even disarmed pathogens.

One way to circumvent this problem is to combine antivirulence compounds with antibiotics to benefit from both virulence suppression and effective pathogen removal [6,11]. While a few studies have already considered such combinatorial treatments [12–18], we currently have no comprehensive understanding of how different types of antibiotics and antivirulence drugs interact, whether interactions are predominantly synergistic or antagonistic, and how combinatorial treatments affect growth and the spread of antibiotic resistant strains. Such knowledge is, however, essential if such therapies are supposed to make their way into the clinics, as drug interactions and their effects on antibiotic resistance evolution will determine both the efficacy and sustainability of treatments. Here, we tackle these open issues by combining four different classes of antibiotics with two antivirulence compounds as treatments against the opportunistic human pathogen *Pseudomonas aeruginosa*, to test the nature of drug interactions and the usefulness of antivirulence compounds as adjuvants to combat antibiotic sensitive and resistant strains.

*P. aeruginosa* is one of the ESKAPE pathogens with multidrug resistant strains spreading worldwide and infections becoming increasingly difficult to treat [19,20]. In addition to its clinical relevance, *P. aeruginosa* has become a model system for antivirulence research [21]. Several antivirulence compounds targeting either the species' quorum sensing (QS) [22–24] or siderophore-mediated iron uptake systems [25–28] have been proposed. While QS is a cell-to-cell communication system that controls the expression of multiple virulence factors, including exo-proteases, biosurfactants, and toxins, siderophores are secondary metabolites important for the scavenging of iron from host tissue. For our experiments, we chose antivirulence compounds that target these two different virulence mechanisms: furanone C-30, an inhibitor of the Las QS-system [12,22], and gallium, targeting the iron-scavenging pyoverdine and iron metabolism [25, 27,29–33].

Furanone C-30 is a synthetic, brominated furanone, which includes the same lactone ring present in the N-acylhomoserine lactone (AHL) QS molecules of *P. aeruginosa* [34]. As a consequence, it can disrupt QS-based communication by competing with the AHL molecules for binding to the main QS (LasR) receptor in a concentration-dependent manner [12,35]. Gallium is an iron mimic whose ionic radius and coordination chemistry is comparable to ferric iron, although its redox properties are different. Specifically, gallium cannot be reduced and thereby irreversibly binds to siderophores and hinders siderophore-mediated iron uptake [25,27]. Pyoverdine is controlled in complex ways involving information on cell density and feedback from iron uptake rates [36,37]. Gallium interferes with these regulatory circuits, whereby cells intermittently up-regulate pyoverdine production at low gallium concentrations to compensate for the increased level of iron limitation, and down-regulate pyoverdine production at higher gallium concentrations because iron uptake via this pathway is unsuccessful [27]. Importantly, gallium drastically reduces pyoverdine availability at the population level in a concentration-dependent manner [25,38].

We combined the two antivirulence compounds with four clinically relevant antibiotics (ciprofloxacin, colistin, meropenem, and tobramycin), which are widely used against *P. aeruginosa* [39]. In a first step, we measured treatment effects on bacterial growth and virulence factor production for all eight drug combinations, for 81 concentration combinations each. In a second step, we applied the Bliss independence model to calculate the degree of synergy or antagonism to obtain comprehensive interaction maps for all combinations both for growth and virulence factor production. Next, we selected for AtbR clones and tested whether the addition of antivirulence compounds as adjuvants restores growth inhibition and affects selection for antibiotic resistance. Finally, we sequenced the genomes of the evolved AtbR clones to understand the genetic basis that might drive the observed effects on pathogen growth and selection for or against antibiotic resistance.

## Results

### Dose-response curves to antibiotics and antivirulence compounds

In a first experiment, we determined the dose-response curve of *P. aeruginosa* PAO1 to each of the four antibiotics (Fig 1) and the two antivirulence compounds (Fig 2) in our experimental media. We found that the dose-response curves for antibiotics followed sigmoid functions (Fig 1) characterized by (i) a low antibiotic concentration range that did not inhibit bacterial growth, (ii) an intermediate antibiotic concentration range that significantly reduced bacterial growth, and (iii) a high antibiotic concentration range that completely stalled bacterial growth.

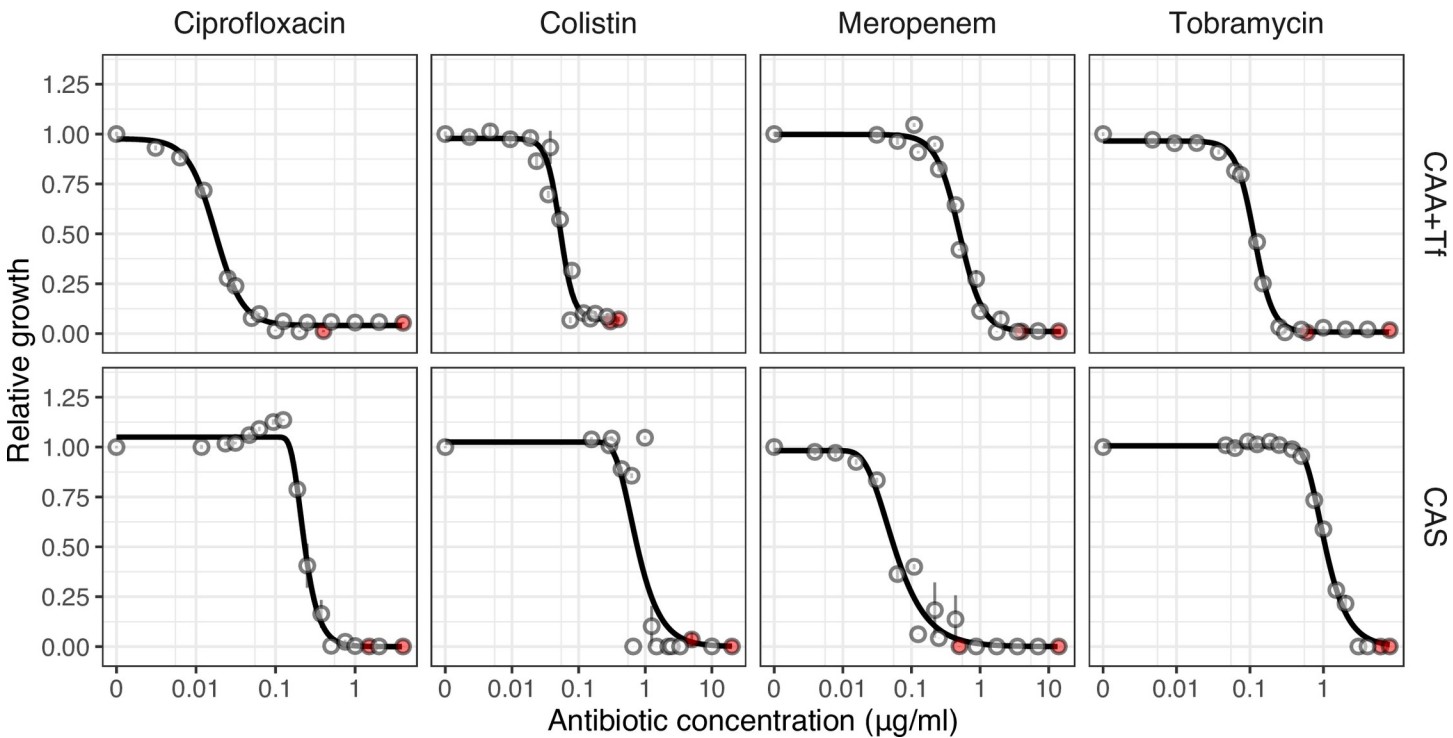

**Fig 1. Antibiotic dose response curves for *P. aeruginosa* PAO1.** We exposed PAO1 to all four antibiotics in two experimental media: CAA+Tf (iron-limited casamino acids medium with transferrin) and CAS (casein medium). Except for meropenem, higher concentrations of antibiotics were required to inhibit PAO1 in CAS compared with CAA+Tf. Dots show means ± standard error across six replicates. All data are scaled relative to the drug-free treatment. Data stem from two independent experiments using different dilution series. The red dots indicate the highest concentration used for the respective experiments, from which 7 serial dilution steps were tested. Curves were fitted with either log-logistic functions (in CAA+Tf) or with three-parameter Weibull functions (in CAS). The underlying data for this figure can be found at https://doi.org/10.6084/m9.figshare.12515364.

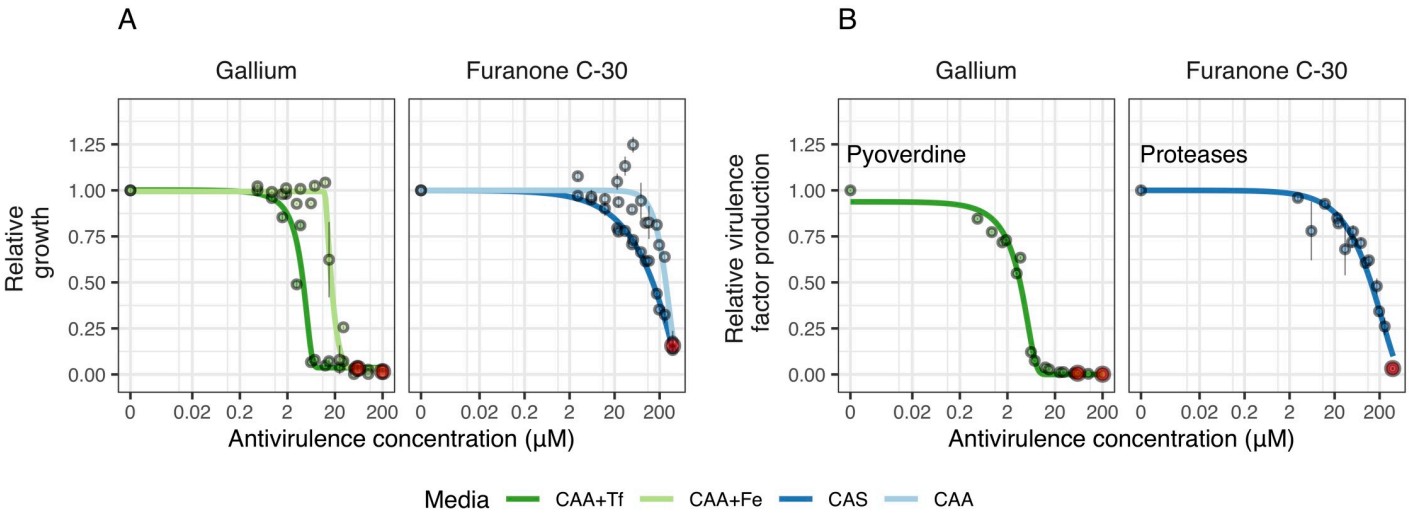

**Fig 2. Antivirulence dose response curves for *P. aeruginosa* PAO1 (growth and virulence factor production).** We exposed PAO1 to the antivirulence compounds gallium (inhibiting pyoverdine-mediated iron uptake) and furanone C-30 (blocking QS response, including protease production) both in media in which the targeted virulence factors are expressed and required (iron-limited CAA+Tf medium for gallium and CAS medium for furanone) and in control media, in which the targeted virulence factors are not required (iron-supplemented CAA+Fe medium for gallium and protein digested CAA for furanone). (**A**) Dose-response curves for growth show that both antivirulence compounds reduced bacterial growth, but more so in media in which the targeted virulence factor is expressed. This demonstrates that there is a concentration window where the antivirulence compounds have no toxic effects on bacterial cells and just limit growth due to virulence factor quenching. (**B**) Dose-response curves for virulence factor production show that gallium and furanone C-30 effectively inhibit pyoverdine and protease production, respectively, in a concentration-dependent manner. Dots show means ± standard errors across six replicates. All data are scaled relative to the drug-free treatment. Data stem from two independent experiments using different dilution series. The red dots indicate the highest concentration used for the respective experiments, from which 7 serial dilution steps were tested. Curves were fitted with either log-logistic functions (in CAA+Tf) or with three-parameter Weibull functions (in CAS). The underlying data for this figure can be found at https://doi.org/10.6084/m9.figshare.12515364. CAA, casamino acid medium; CAS, casein medium; QS, quorum sensing; Tf, human apo-transferrin.

As antibiotics curbed growth, they congruently also reduced the availability of virulence factors at the population level (S1 Fig).

Similarly shaped dose-response curves for growth (Fig 2A) and virulence factor production (Fig 2B) were obtained for gallium (quenching pyoverdine) and furanone C-30 (inhibiting protease production) in the respective media in which the two virulence factors are important for growth. Under such conditions, the reduced availability of virulence factors immediately feeds back on growth by inducing iron starvation (gallium) or the inability to degrade proteins (furanone). Crucially, the dose-response curves for growth shifted to the right (extending phase (i)) when we repeated the experiment in media, where the virulence factors are not needed for growth (i.e., iron-rich media for pyoverdine, and protein digest media for proteases). This shows that there is a window of concentrations where growth inhibition is caused by virulence factor quenching alone. Conversely, high concentrations of antivirulence compounds seem to have additional off-target effects reducing growth.

## Interaction maps of antibiotic-antivirulence drug combinations

**General patterns.** From the dose-response curves, we chose 9 concentrations for each drug to cover the entire trajectory, from no to intermediate to high growth inhibition. We then combined antibiotics with antivirulence compounds in a 9×9 concentration matrix and measured the dose-response curve for every single drug combination for both growth and virulence factor production (Fig 3). At the qualitative level, independent drug effects would cause a symmetrical downshift of the dose-response curve, with higher antivirulence compound concentrations supplemented. We indeed noticed symmetrical downshifts for many dose-

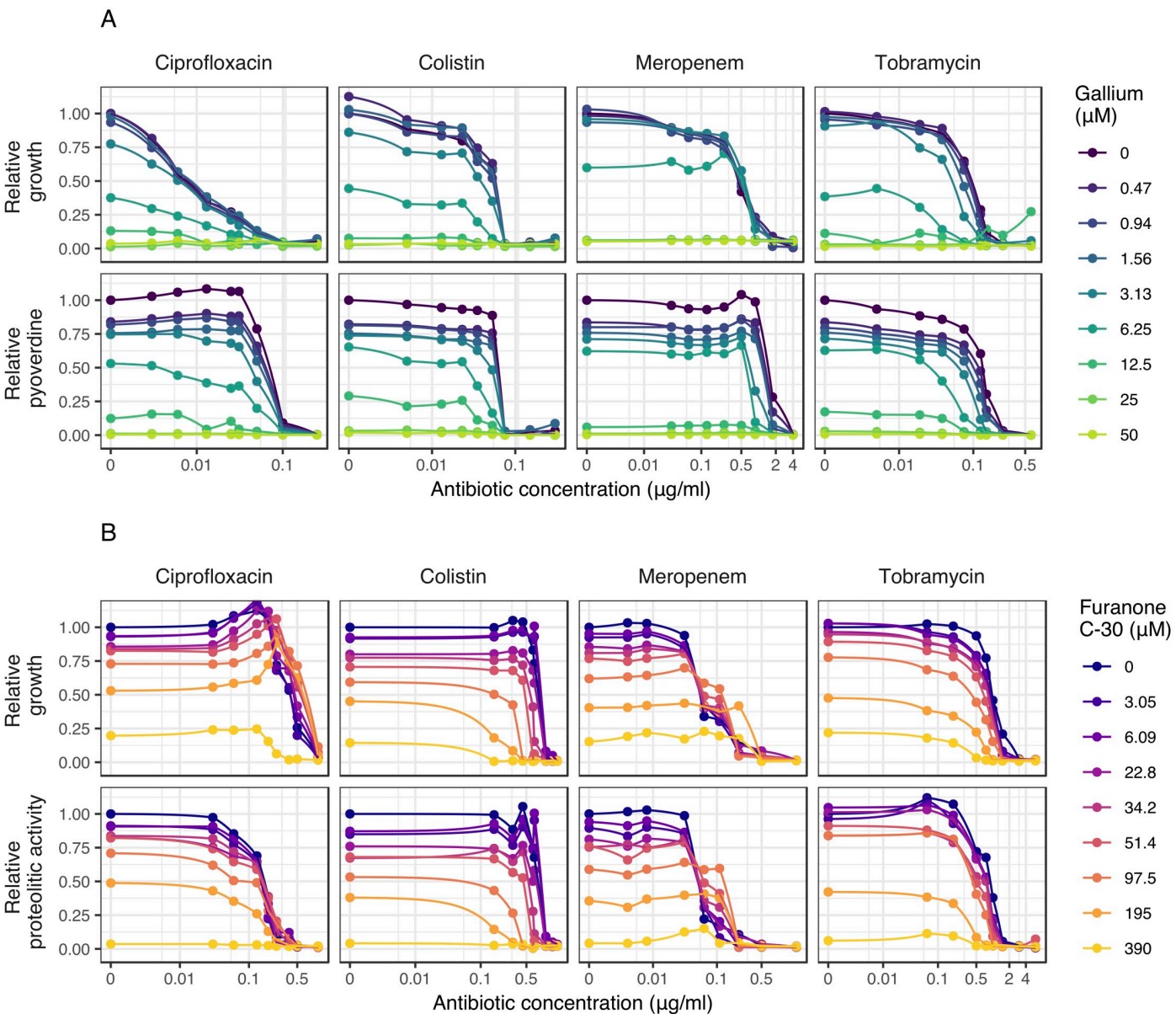

**Fig 3. Dose-response curves for *P. aeruginosa* PAO1 under antibiotic-antivirulence combination treatments.** Dose-response curves for growth and virulence factor production for PAO1 were assessed for 9×9 drug concentration matrixes involving the four antibiotics combined with either gallium (**A**) or furanone C-30 (**B**). Experiments were carried out in media in which the corresponding virulence factors are required for growth (pyoverdine: CAA+Tf; protease: CAS). Growth and virulence factor production were measured after 48 hours. All values are scaled relative to the untreated control, and data points show the mean across 12 replicates from two independent experiments. We used spline functions to fit the dose-response curves. The underlying data for this figure can be found at https://doi.org/10.6084/m9.figshare.12515364. CAA, casamino acid medium; CAS, casein medium; Tf, human apo-transferrin.

response curves (Fig 3), but there were also clear cases of nonsymmetrical shifts, indicating synergy or antagonism between drugs. We then used the Bliss model, representing the adequate model for drugs with independent modes of actions, to quantify these effects. We found patterns of synergy and antagonism for both growth and virulence factor inhibition across the concentration matrices for all drug combinations (Fig 4), with many of these interactions being significant (S2 Fig).

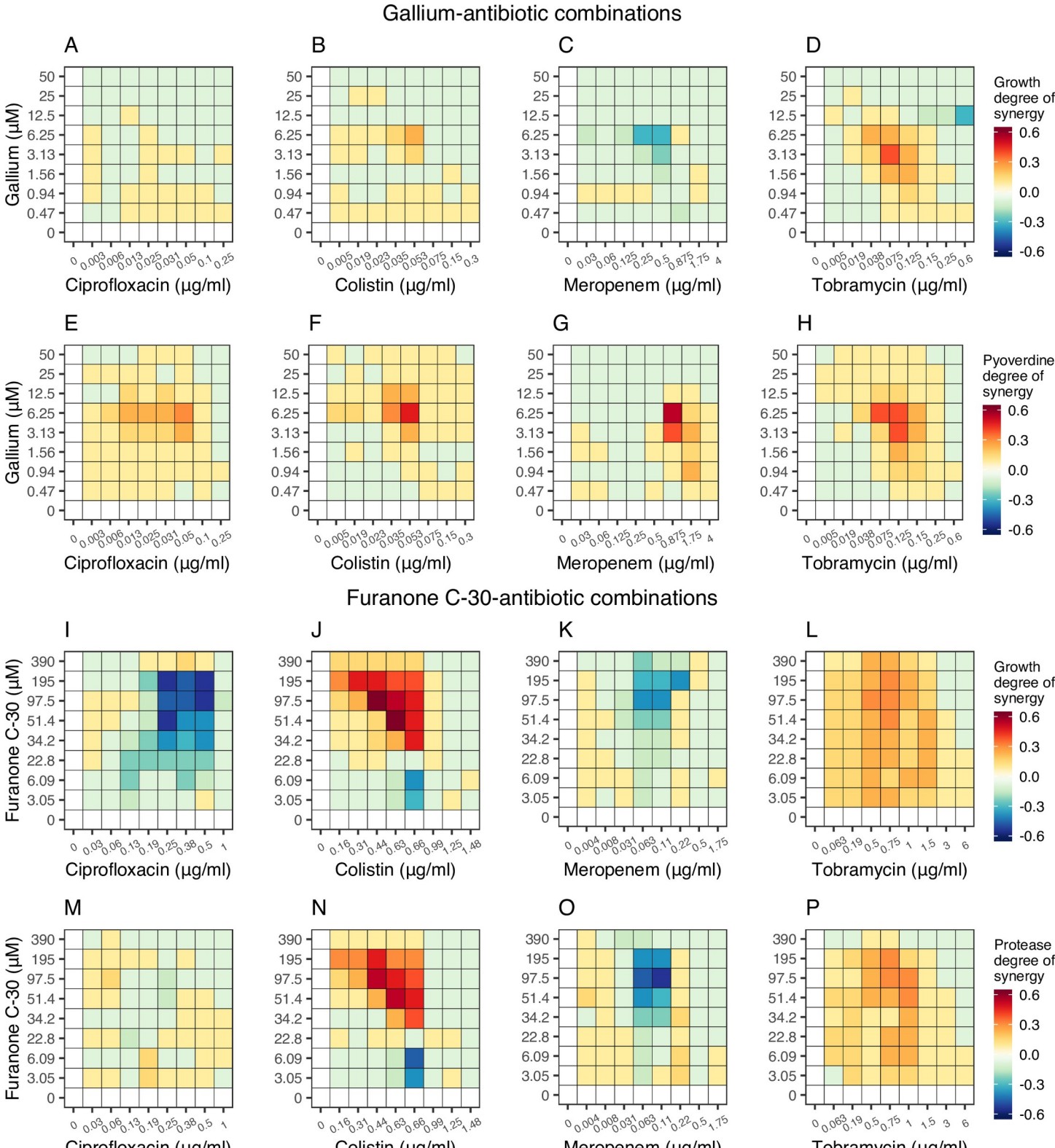

**Fig 4. Drug interaction heatmaps for antibiotic-antivirulence combination treatments.** We used the Bliss independence model to calculate the degree of synergy for every single drug combination with regard to growth suppression and virulence factor quenching shown in Fig 3. Heatmaps depicting variation in drug interactions ranging from antagonism (blue) to synergy (red) are shown for gallium-antibiotic combinations (**A-D** for growth; **E-H** for pyoverdine production) and furanone-

antibiotic combinations (**I-L** for growth; **M-P** for protease production). All calculations are based on 12 replicates from two independent experiments. The underlying data for this figure can be found at https://doi.org/10.6084/m9.figshare.12515364.

**Gallium-antibiotic combinations.** Gallium combined with ciprofloxacin or colistin had mostly independent effects on bacterial growth (i.e., weak or no synergy/antagonism) (Fig 4A and 4B). With regard to the inhibition of pyoverdine production, both drug combinations showed significant levels of synergy at intermediate drug concentrations (Fig 4E and 4F, S2 Fig). For gallium-meropenem combinations, we observed mostly independent interactions for growth and pyoverdine inhibition, with small hotspots of antagonism (for growth) and synergy (for siderophore inhibition) existing at intermediate drug concentrations (Fig 4C and 4G). Finally, for gallium-tobramycin combinations, there were relatively strong significant synergistic interactions for both growth and pyoverdine inhibition at intermediate drug concentrations (Fig 4D and 4H, S2 Fig). Interestingly, we observed synergy with regard to pyoverdine inhibition for all drug combinations, indicating that the combination of low cell density induced by the antibiotics and gallium-mediated pyoverdine quenching is a successful strategy to repress this virulence factor.

**Furanone-antibiotic combinations.** For furanone-ciprofloxacin combinations, we found relatively strong significant antagonistic interactions with regard to growth inhibition (Fig 4I), whereas effects on protease inhibition were mostly independent (Fig 4M). In contrast, for furanone-colistin combinations we observed strong and significant synergistic drug interactions, especially for intermediate and higher concentrations of the antivirulence compound for growth and protease inhibition (Fig 4J and 4N, S2 Fig). Furanone-meropenem, on the other hand, interacted mostly antagonistically with regard to growth and protease inhibition (Fig 4K and 4O). Conversely, for furanone-tobramycin combinations, there were pervasive significant patterns of synergy across the entire drug combination range for growth and virulence factor inhibition (Fig 4L and 4P, S2 Fig).

## Do the degrees of synergy for growth and virulence factor inhibition correlate?

As the combination treatments affect both growth and virulence factor production, we examined whether the degrees of synergy correlate between the two traits (S3 Fig). For gallium-antibiotic combinations, we found no correlations for ciprofloxacin and meropenem, but positive associations for colistin and tobramycin (Pearson correlation coefficient; ciprofloxacin: $r = 0.09$, $t_{79} = 0.85$, $p = 0.394$; colistin: $r = 0.69$, $t_{79} = 8.51$, $p < 0.001$; meropenem: $r = 0.17$, $t_{79} = 1.53$, $p = 0.130$; tobramycin: $r = 0.58$, $t_{79} = 6.39$, $p < 0.001$). For furanone-antibiotic combinations, there were strong positive correlations between the levels of synergy for the two traits for all drug combinations (ciprofloxacin: $r = 0.34$, $t_{79} = 3.22$, $p = 0.002$; colistin: $r = 0.96$, $t_{79} = 32.50$, $p < 0.001$; meropenem: $r = 0.87$, $t_{79} = 15.48$, $p < 0.001$; tobramycin: $r = 0.75$, $t_{79} = 10.16$, $p < 0.001$).

## Antibiotic resistance can lead to collateral sensitivity and cross-resistance to antivirulence compounds

In a next step, we asked whether antivirulence compounds could be used as adjuvants to suppress the growth of antibiotic resistant (AtbR) clones. To address this question, we first experimentally selected, isolated, and analyzed AtbR clones. We aimed for one clone per antibiotic and medium. In the end, we examined seven clones, as only one selection line survived the ciprofloxacin selection regime (see Materials and methods for details). We then assessed the dose-

response curve of these clones for the respective antibiotics to confirm and quantify their level of resistance (S4 Fig). We further established the dose-response curves of all AtbR clones for the two antivirulence compounds to test for collateral sensitivity and cross-resistance. Here, we compared the half maximal inhibitory concentration ($IC_{50}$) values between the AtbR and wild-type (WT) strain (S5 Fig), and found evidence for weak but significant collateral sensitivity between colistin and gallium and relatively strong collateral sensitivity between tobramycin and furanone. Conversely, we found that resistance to ciprofloxacin, colistin, and also to some extent to meropenem can confer cross-resistance to furanone (all statistical analyses are shown in S5 Fig).

Based on these experiments, we picked two concentrations for gallium (low, 1.56 μM; intermediate, 6.25 μM) and furanone (low, 6.3 μM; intermediate, 22.8 μM) as adjuvants in combination with antibiotics to test whether antivirulence compounds can restore growth inhibition of and alter selection for AtbR clones.

## Antivirulence compounds as adjuvants can restore growth inhibition of antibiotic resistant strains

In a first set of experiments, we subjected all AtbR clones and the antibiotic sensitive WT to antivirulence treatments alone and to combination treatments with antibiotics (Fig 5). For gallium, we observed that the WT and AtbR clones responded almost identically to the antivirulence compounds in the absence of antibiotics, showing that AtbR clones are still sensitive to gallium (Fig 5A). When treated with antibiotics, we observed that the addition of antivirulence compounds consistently reduced growth of all AtbR clones (Fig 5A). The level of synergy

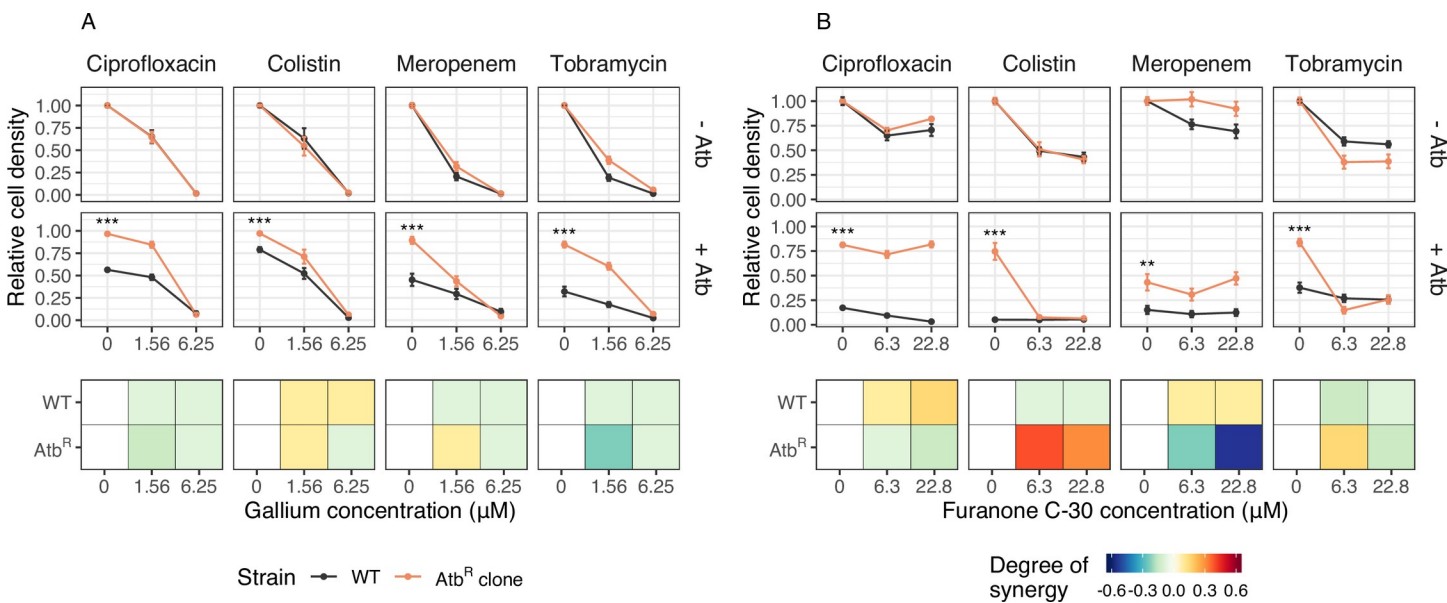

**Fig 5. Effect of combination treatment on growth of AtbR clones.** Test of whether the addition of gallium (**A**) or furanone (**B**) can restore growth suppression in AtbR clones (in orange) relative to the susceptible wild-type (WT; in black). Under antibiotic treatment and in the absence of antivirulence compounds, all AtbR clones grew significantly better than the WT (two-sample $t$ tests, $-26.63 \leq t_{21-40} \leq -3.03$, $p < 0.01$ for all treatment combinations; n.s. = nonsignificant; $^{*}p < 0.05$; $^{**}p < 0.01$; $^{***}p < 0.001$), a result that holds for both scaled (as shown above) and absolute growth. In the presence of antivirulence compounds (upper series of panels without antibiotics [−Atb]; lower series of panels with antibiotics [+Atb]), growth suppression was restored in six out of eight cases. The exceptions were the ciprofloxacin-furanone and meropenem-furanone combinations. The bottom series of panels shows the degree of drug synergy for the WT and the AtbR clones. All cell density values (measured with flow cytometry as number of events detected in 5 μL of culture, after 24 hours) are scaled relative to the untreated control. All data are shown as means ± standard errors across a minimum of 16 replicates from 4 to 6 independent experiments. The underlying data for this figure can be found at https://doi.org/10.6084/m9.figshare.12515364. AtbR clones, antibiotic resistant clones.

between drugs is similar for WT and AtbR clones, with the effects being close to zero (varying between weak antagonism and weak synergy) in most cases. Altogether, these findings indicate that gallium acts independently to all the tested antibiotics and is still able to induce iron starvation and thus to reduce growth in AtbR clones. Important to note is that gallium has a stronger growth inhibitory effect in this experiment compared to the dose-response curve analysis (S5 Fig, collected after 48 hours), because this experiment ran for 24 hours only.

For furanone, the patterns were more diverse (Fig 5B). There were two cases of full cross-resistance (ciprofloxacin and meropenem), in which the addition of furanone no longer had any effect on bacterial growth. In these cases, we observed a change from weak drug synergy (for the WT) to strong drug antagonism (for the AtbR clones). In contrast, we found a strong shift from weak antagonism (for the WT) to strong drug synergy (for the AtbR clone) when furanone was combined with colistin. In this case, furanone re-potentiated the antibiotic. Note that we initially also observed a certain level of cross-resistance for this drug combination (S5 Fig), however, only at much higher furanone concentrations than those used here. Finally, the pattern between tobramycin and furanone was driven by collateral sensitivity, where the addition of furanone to the AtbR clone completely restored growth inhibition.

## Antivirulence as adjuvants can abrogate or reverse selection for antibiotic resistance

We then investigated whether antivirulence compounds alone or in combination with antibiotics can influence the spread of AtbR clones in mixed populations with susceptible WT cells (Fig 6). First, we competed the AtbR clones against the susceptible WT in the absence of any treatment and observed that AtbR clones consistently lost the competitions (one-sample $t$ tests, $-13.50 \leq t_{15-26} \leq -2.62$, $p \leq 0.050$ for all comparisons). This confirms that antibiotic resistance is costly and is selected against in the absence of treatment. We then added the antivirulence drug alone using the same concentrations as for the monoculture experiments (Fig 5). We found that the AtbR clones did not experience a selective advantage in 14 out 16 cases, with the exception being one colistin resistant clone that slightly increased in frequency (Fig 6). This analysis shows that the cost of AtbR resistance is largely maintained in the presence of antivirulence compounds. Next, we exposed the mixed cultures to the antibiotics alone and observed that, as expected, AtbR clones always experienced a significant fitness advantage under treatment (one-sample $t$ test, $4.54 \leq t_{19-26} \leq 13.41$, $p < 0.001$ for all combinations).

When combining antivirulence compounds with antibiotics, three different relative fitness patterns emerged for AtbR clones. In three cases (colistin-gallium, ciprofloxacin-furanone, and meropenem-furanone), AtbR clones experienced large fitness advantages and were selectively favored regardless of whether the antivirulence compound was present or not. In four cases (ciprofloxacin-gallium, meropenem-gallium, tobramycin-gallium, colistin-furanone), the addition of antivirulence compounds gradually reduced the relative fitness of the AtbR clones, whereby in two cases (meropenem-gallium, tobramycin-gallium) the selective advantage of AtbR clones was completely abrogated. Finally, in one case (tobramycin-furanon) selection for AtbR clones was even reversed and AtbR clones lost the competition.

## Drug synergy does not predict selection against antibiotic resistance

We examined whether drug interactions, ranging from antagonism to synergy for both AtbR clones and the WT (Fig 5) correlate with their relative fitness in competition under combination treatments. However, we found no support for such associations (S6 Fig, ANOVA, AtbR: $F_{1,65} = 0.88$, $p = 0.353$; WT: $F_{1,65} = 1.85$, $p = 0.179$), but instead observed that variation in

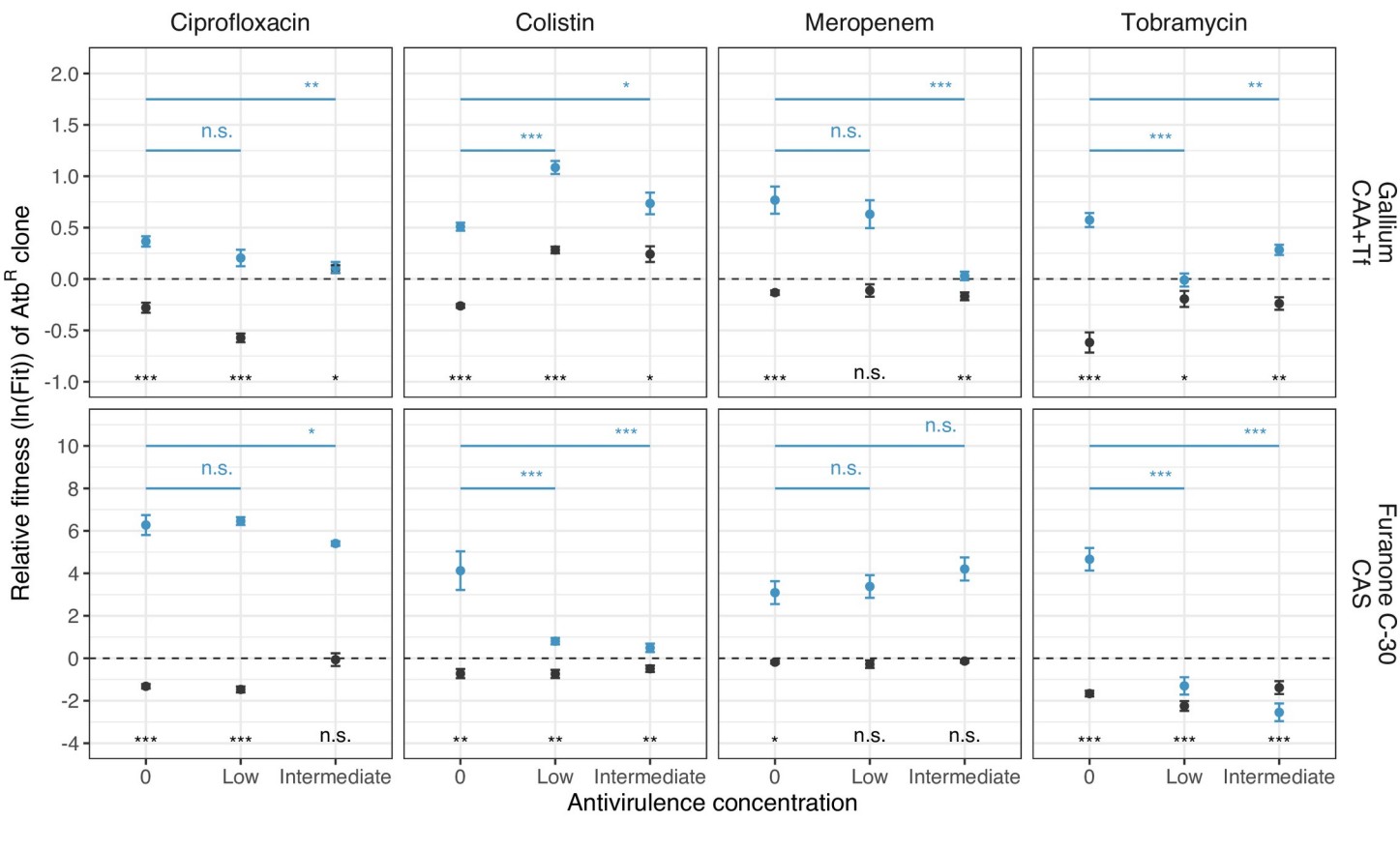

**Fig 6. Effect of combination treatment on the relative fitness of AtbR clones.** Test of whether antivirulence compounds alone or in combination with antibiotics can abrogate or revert selection for antibiotic resistance. All AtbR clones were competed against the susceptible WT for 24 hours, starting at a 1:9 ratio. The dashed lines denote fitness parity, where none of the competing strains has a fitness advantage. In the absence of any treatment, all AtbR clones showed a fitness disadvantage (fitness values < 0) compared to the WT, demonstrating the cost of resistance. When treated with antivirulence compounds alone, the AtbR clones did not experience any selective advantage in 14 out of 16 cases (exception: colistin-gallium combinations). When treated with antibiotics alone, all AtbR clones experienced significant fitness advantages (fitness values > 0), as expected. When antivirulence compounds were added as adjuvants to antibiotics, the fitness advantage of AtbR clones was reduced, abrogated, or reversed for six out of eight drug combinations. All data are shown as means ± standard errors across a minimum of 16 replicates from 4 to 7 independent experiments. Significance levels are based on *t* tests or ANOVAs: n.s. = non-significant; $^*p < 0.05$; $^{**}p < 0.01$; $^{***}p < 0.001$. See S1 Table for full details on the statistical analyses. The underlying data for this figure can be found at https://doi.org/10.6084/m9.figshare.12515364. - Atb, treatments without antibiotics; + Atb, treatments with antibiotics; AtbR clones, antibiotic resistant clones; CAA, casamino acid medium; CAS, casein medium; Tf, human apo-transferrin; WT, wild-type.

fitness patterns was explained by specific drug combinations (antivirulence-antibiotic interaction, AtbR: $F_{3,65} = 37.45$, $p < 0.001$, WT: $F_{3,65} = 14.50$, $p < 0.001$).

## Genetic basis of experimentally evolved antibiotic resistance

The whole-genome sequencing of the experimentally evolved AtbR clones revealed a small number of single nucleotide polymorphisms (SNPs) and insertions or deletions (INDELs), which are known to be associated with resistance to the respective antibiotics (Table 1). The AtbR clone resistant to ciprofloxacin had mutations in *gyrB*, a gene encoding the DNA gyrase subunit B, the direct target of the antibiotic [40]. In addition, we identified an 18-bp deletion in the *mexR* gene, encoding a multidrug efflux pump repressor [41]. The two AtbR clones resistant to colistin had different mutations in the same target gene *phoQ* (a non-synonymous SNP in one clone versus a 1-bp insertion in addition to a non-synonymous SNP in the other clone). PhoQ is a regulator of the lipopolysaccharide modification operon, and mutations in

**Table 1. List of mutations in the AtbR clones.**

| AtbR clone | Combination with | Gene[a] | Description | Mutation type | Reference | Variant | Position[b] | Reference |
|---|---|---|---|---|---|---|---|---|
| CpR_1 | Gallium Furanone C-30 | gyrB | DNA gyrase subunit B | INDEL | CCCAGGAG | CG | 5,671 | [40] |
| | | mexR | Multidrug resistance operon repressor | INDEL | ATCAGTGCCTTGTCGCGGCA | AA | 471,547 | [50,51] |
| CoR_1 | Gallium | phoQ | Two-component sensor PhoQ | SNP | T | G | 1,279,140 | [42,52–54] |
| CoR_2 | Furanone C-30 | phoQ | Two-component sensor PhoQ | INDEL | T | TC | 1,279,085 | [42,52–54] |
| | | | | SNP | T | C | 1,279,089 | |
| | | PA1327 | Probable protease | INDEL | CA | C | 1,440,622 | |
| MeR_1 | Gallium | mpl | UDP-N-acetylmuramate:L-alanyl-gamma-D-glutamyl- meso-diaminopimelate ligase | SNP | T | G | 4,499,740 | [43,55] |
| MeR_2 | Furanone C-30 | parR | Two-component response regulator, ParR | SNP | C | T | 1,952,257 | [44] |
| | | PA1874 | Hypothetical protein (efflux pump) | COMPLEX | ACCGGG | CCCGTC | 2,039,326 | [45] |
| | | | | SNP | A | G | 2,039,485 | |
| | | | | SNP | C | T | 2,039,494 | |
| | | | | SNP | T | C | 2,039,512 | |
| | | | | SNP | G | A | 2,039,517 | |
| | | | | SNP | T | C | 2,039,524 | |
| | | | | COMPLEX | CGGG | TGGC | 2,039,530 | |
| | | nalD | Transcriptional regulator NalD | SNP | A | C | 4,006,981 | [46,47] |
| TbR_1 | Gallium | fusA1 | Elongation factor G | SNP | C | T | 4,769,121 | [48] |
| | | | | SNP | T | G | 5,253,694 | |
| TbR_2 | Furanone C-30 | fusA1 | Elongation factor G | SNP | T | C | 4,770,785 | [48] |
| | | pscP | Translocation protein in type III secretion system | INDEL | TG(TTGGCG)$_{\times 11}$ | TG (TTGGCG)$_{\times 12}$ | 1,844,903 | |
| | | PA4132 | Conserved hypothetical protein | SNP | A | C | 4,621,443 | |

[a]Mutations in the ancestor WT background compared with the reference *P. aeruginosa* PAO1 genome are reported in [38].

[b]Position on the *P. aeruginosa* PAO1 reference genome [56].

Abbreviations: Atb, antibiotic; AtbR clones, antibiotic resistant clones; Co, colistin; COMPLEX, multiple consecutive SNPs; Cp, ciprofloxacin; INDEL, insertion or deletion; Me, meropenem; SNP, single nucleotide polymorphism; Tb, tobramycin; WT, wild-type

this gene represent the first step in the development of high-level colistin resistance [42]. One AtbR clone resistant to meropenem had a non-synonymous SNP in the coding sequence of *mpl*. This gene encodes a murein tripeptide ligase, which contributes to the overexpression of the beta-lactamase precursor gene *ampC* [43]. The other AtbR clone resistant to meropenem had mutations in three different genes, which can all be linked to antibiotic resistance mechanisms: we found (i) one non-synonymous SNP in *parR*, which encodes a two-component response regulator involved in several resistance mechanisms, including drug efflux, porin loss, and lipopolysaccharide modification [44]; (ii) 7 mutations in the *PA1874* gene, which encodes an efflux pump [45]; and (iii) one non-synonymous SNP in *nalD*, encoding the transcriptional regulator NalD, which regulates the expression of drug-efflux systems [46,47]. Both AtbR clones resistant to tobramycin had non-synonymous SNPs in *fusA1*. This gene encodes the elongation factor G, a key component of the translational machinery. Although aminoglycosides do not directly bind to the elongation factor G and the complete resistance mechanism

is still unknown, mutations in *fusA1* are associated with high resistance to tobramycin and are often found in clinical isolates [48,49].

## Discussion

In this study, we systematically explored the effects of combining antibiotics with antivirulence compounds as a potentially promising strategy to fight susceptible and antibiotic resistant opportunistic human pathogens. Specifically, we combined four different antibiotics (ciprofloxacin, colistin, meropenem, tobramycin) with two antivirulence compounds (gallium targeting siderophore-mediated iron uptake and furanone C-30 targeting the QS communication system) in 9×9 drug interaction matrices against the bacterium *P. aeruginosa* as a model pathogen. Our heatmaps reveal drug combination–specific interaction patterns. While colistin and tobramycin primarily interacted synergistically with the antivirulence compounds, independent and antagonistic interactions occurred for ciprofloxacin and meropenem in combination with the antivirulence compounds (Figs 3 and 4). We then used antivirulence compounds as adjuvants and observed that they can restore growth inhibition of AtbR clones in six out of eight cases (Fig 5). Finally, we performed competition assays between antibiotic resistant and susceptible strains under single and combinatorial drug treatments and found that antivirulence compounds can reduce (two cases), abrogate (two cases), or even reverse (one case) selection for antibiotic resistance (Fig 6).

Our results identify antibiotic-antivirulence combinations as a potentially powerful tool to efficiently treat infections of troublesome nosocomial pathogens such as *P. aeruginosa*. From the eight combinations analyzed, tobramycin-antivirulence combinations emerged as the top candidate treatments because (i) drugs interacted synergistically both with regard to growth and virulence factor inhibition; (ii) the antivirulence compounds restored growth inhibition of AtbR clones, and even re-potentiated tobramycin in the case of furanone; and (iii) antivirulence compounds either abrogated (for gallium) or even reversed (for furanone) selection for tobramycin resistance, due to collateral sensitivity in the latter case. Meropenem-gallium emerged as an additional promising combination, as this combination also restored growth inhibition of the AtbR clone and abrogated its selective advantage, despite this drug combination showing slight but significant antagonistic effects.

Drug synergy is desirable from a clinical perspective because it allows using lower drug concentrations, thereby minimizing side effects while maintaining treatment efficacy [57,58]. In this context, a number of studies have examined combinations of antibiotics and antivirulence compounds targeting various virulence factors including QS, iron uptake, and biofilm formation in *P. aeruginosa* [12,13, 15–18, 23,59,60]. While some of these studies have used a few specific concentration combinations to qualitatively assess synergy, we here present comprehensive quantitative interaction maps for these two classes of drugs. A key insight of these interaction maps is that specific drug combinations cannot simply be classified as either synergistic or antagonistic. Instead, drug interactions are concentration dependent, with most parts of the interaction maps being characterized by independent effects interspersed with hotspots of synergy or antagonism (Fig 4 and S2 Fig). The strongest effects of synergy and antagonism are often observed at intermediate drug concentrations, which are, in the case of synergy, ideal for developing combinatorial therapies that maximize treatment efficacy while minimizing toxicity for the patient. A crucial next step would be to test whether the same type of drug interactions can be recovered in animal host models. Although one should be careful to relate absolute concentrations from an in vitro assay to actual drug concentrations administered to patients and clinical antibiotic breakpoints [61], we found that the dose-responses of our PAO1 WT strain (and the resulting minimal inhibitory concentrations [MICs], Fig 1) are

close to the average MIC values (MIC50) reported in a study across 7,452 clinical *P. aeruginosa* isolates [62]. This means that the drug synergies observed in our study might operate at a range that is also relevant for clinical isolates.

While drug antagonism is considered undesirable from a clinical perspective, work on antibiotic combination therapies has revealed that antagonistic interactions can inhibit the spread of antibiotic resistance [63–65]. The reason behind this phenomenon is that when two drugs antagonize each other, becoming resistant to one drug will remove the antagonistic effect on the second drug, such that the combination treatment will be more effective against the resistant clones [63]. We suspected that such effects might also occur for antagonistic antibiotic-antivirulence treatments, and indeed the meropenem-gallium combination discussed above matches this pattern. However, when comparing across all the eight combinations, we found no evidence that the selection for or against AtbR clones correlates with the type of drug interaction (S6 Fig). A possible explanation for the overall absence of an association is that the antagonism between antibiotics and antivirulence compounds was quite moderate. In contrast, previous work used an extreme case of antagonism, where the effect of one drug was almost completely suppressed in the presence of the second drug [63,65].

We propose that it is rather the underlying molecular mechanism and not the direction of drug interaction that determines whether selection for antibiotic resistance is compromised or maintained. For instance, any resistance mechanism that reduces antibiotic entry or increases its efflux could conceivably confer cross-resistance to antivirulence compounds, which should in turn maintain or even potentiate and not reverse selection for antibiotic resistance. This phenomenon could explain the patterns observed for furanone in combination with ciprofloxacin and meropenem, where in both cases our sequencing analysis revealed mutations in genes regulating efflux pumps (Table 1). Because furanone needs to enter the cells to become active, these mutations, known to confer resistance to antibiotics [42, 45,50], likely also confer resistance to furanone [66]. In contrast, efflux pumps up-regulation cannot work as a cross-resistance mechanism against gallium, which binds to secreted pyoverdine and thus acts outside the cell [38].

Alternatively, competitive interactions between resistant and sensitive pathogens over common resources could compromise the spread of drug resistance, as shown for malaria parasites [67]. In our case, it is plausible to assume that AtbR clones are healthier than susceptible cells and might therefore produce higher amounts of pyoverdine and proteases under antivirulence treatment. Because these virulence factors are secreted and shared between cells, AtbR clones take on the role of cooperators: they produce costly virulence factors that are then shared with and exploited by the susceptible cells [27,68,69]. This scenario could apply to tobramycin-gallium/furanone combinations, where resistant clones had mutations in *fusA1* known to be associated with the restoration of protein synthesis [48]. Similar social effects could explain selection abrogation in the case of meropenem-gallium combination. Here, the meropenem resistant clone has a mutation in *mpl*, which can trigger the overexpression of the β-lactamase *ampC* resistance mechanism [43]. Because β-lactamase enzyme secretion and extracellular antibiotic degradation is itself a cooperative behavior [70], it could, together with the virulence factor sharing described above, compromise the spread of the resistant clone. Clearly, all these explanations remain speculative and further studies are required to understand the molecular and evolutionary basis of abrogated and reversed selection for resistance.

In summary, drug combination therapies are gaining increased attention as more sustainable strategies to treat infections, limiting the spread of antibiotic resistance [71–74]. They are already applied to a number of diseases, including cancer [75], HIV [76], and tuberculosis infections [77]. Here, we probed the efficacy and evolutionary robustness of antibiotics combined with antivirulence compounds. This is an interesting combination because antibiotic

treatments alone face the problem of rapid resistance evolution, whereas antivirulence drugs are evolutionarily more robust but can only disarm and not eliminate pathogens. Combinatorial treatments seem to bring the strengths of the two approaches together: efficient removal of bacteria by the antibiotics combined with disarming and increased evolutionary robustness of the antivirulence compounds. While our findings are promising and could set the stage for a novel class of combinatorial treatments, there are still many steps to take to bring our approach to the clinics. First, it would be important to quantify the rate of resistance evolution directly under the combinatorial treatments to test whether drug combination itself slows down resistance evolution [71]. The level of evolutionary robustness of the antivirulence compound would be of particular importance here, as it is known to vary across compounds [27,38]. For example, previous studies showed that resistance to furanone can arise relatively easily [66], while gallium seems to be more evolutionarily robust [38]. Second, the various antibiotic-antivirulence combinations must be tested in relevant animal host models, as host conditions including the increased spatial structure inside the body can affect the competitive dynamics between strains, potentially influencing the outcome of the therapy [78,79]. Relevant in this context would also be tests that closely examine the toxicity of the antivirulence compounds for mammalian cells, an aspect that has received little attention so far [17,80]. Third, our findings suggest that the beneficial effects of combination therapy depend on the specific antibiotic resistance mechanism involved. This hypothesis should be tested in more detail by using sets of mutants that are resistant to the same antibiotic but through different mechanisms. Finally, the observed patterns of drug synergy (Fig 4) and reversed selection for resistance (Fig 6) are concentration dependent. Thus, detailed research on drug delivery and the pharmacodynamics and pharmacokinetics of combination therapies would be required [65], especially to determine the drug interaction patterns within patients.

## Materials and methods

### Bacterial strains

For all our experiments, we used *P. aeruginosa* PAO1 (ATCC 15692). In addition to the WT PAO1, we further used two isogenic variants tagged with either a constitutively expressed GFP (green fluorescent protein) or mCherry (red fluorescent protein). Both fluorescently tagged strains were directly obtained from the WT PAO1 using the miniTn7-system to chromosomally integrate a single stable copy of the marker gene, under the strong constitutive Ptac promoter, at the *att*Tn7 site [81]. The gentamycin resistance cassette, required to select for transformed clones, was subsequently removed using the pFLP2-encoded recombinase [81]. AtbR clones used for competition assays were generated through experimental evolution and are listed in Table 1 together with their respective mutations.

### Media and growth conditions

For all experiments, overnight cultures were grown in 8 mL of Lysogeny broth (LB) in 50-mL Falcon tubes, incubated at 37°C, 220 rpm for 18 hours. We washed overnight cultures with 0.8% NaCl solution and adjusted them to optical density at 600 nm ($OD_{600}$) = 1. Bacteria were further diluted to a final starting $OD_{600} = 10^{-3}$ for all experiments. We used two different media, in which the targeted virulence factors (pyoverdine or protease) are important. For pyoverdine, we used iron-limited casamino acid medium (CAA) (CAA+Tf) (0.5% casamino acids, 5 mM $K_2HPO_4 \cdot 3H_2O$, 1 mM $MgSO_4 \cdot 7H_2O$), buffered at neutral pH with 25 mM HEPES buffer and supplemented with 100 μg/mL human apo-transferrin to chelate iron and 20 mM $NaHCO_3$ as a cofactor. As an iron-rich control medium, we used CAA supplemented

with 25 mM HEPES and 20 μM $FeCl_3$, but without apo-transferrin and 20 mM $NaHCO_3$ to create conditions that do not require pyoverdine for growth [36].

For QS-regulated proteases, we used casein medium (CAS) (0.5% casein, 5 mM $K_2HPO_4 \cdot 3H_2O$, 1 mM $MgSO_4 \cdot 7H_2O$) supplemented with 25 mM HEPES buffer and 0.05% CAA. In this medium, proteases are required to digest the casein. A small amount of CAA was added to allow cultures to have a growth kick start prior to protease secretion [82]. As a control, we used CAA supplemented with 25 mM HEPES buffer, a medium in which proteases are not required. All chemicals were purchased from Sigma-Aldrich, Buchs SG, Switzerland. The CAS medium is intrinsically turbid due to the poor solubility of casein, which can interfere with the growth kinetics measured via optical density (S7A Fig). To solve this issue, we used mCherry fluorescence intensity as a reliable proxy for growth in CAS (S7B and S7C Fig).

### Single drug growth and virulence factor inhibition curves

To determine the activity range of each antibiotic (ciprofloxacin, colistin, meropenem, and tobramycin) and antivirulence drug (gallium as $GaNO_3$ and furanone C-30), we subjected PAO1 bacterial cultures to two different 7-step serial dilutions for each antibacterial. Ciprofloxacin: 0–4 μg/mL; colistin: 0–0.4 μg/mL in CAA+Tf and 0–20 μg/mL in CAS; meropenem: 0–14 μg/mL; tobramycin: 0–8 μg/mL; gallium: 0–200 μM; furanone C-30: 0–390 μM. All antibacterials were purchased from Sigma-Aldrich, Buchs SG, Switzerland. Overnight cultures were prepared and diluted as explained above and then added into 200 μL of media on 96-well plates with six replicates for each drug concentration. Plates were incubated statically at 37˚C, and growth was measured either as $OD_{600}$ (in CAA+Tf) or mCherry fluorescence (excitation 582 nm, emission 620 nm in CAS) after 48 hours using a Tecan Infinite M-200 plate reader (Tecan Group, Männedorf, Switzerland). Control experiments (S8 Fig) confirmed that endpoint $OD_{600}$ or mCherry measurements showed strong linear correlations ($0.858 < R^2 < 0.987$) with the growth integral (area under the growth curve), which is a good descriptor of the overall inhibitory effects covering the entire growth period [83].

At this time point, we further quantified pyoverdine production through its natural fluorescence (excitation 400 nm, emission 460 nm, a readout that scales linearly with pyoverdine concentration in the medium) and protease production in the cell-free supernatant using the protease azocasein assay (adapted from [38], shortening incubation time to 30 minutes). The two metals, gallium and bromine (in Furanone C-30), alter the fluorescence levels of pyoverdine and mCherry in a concentration-dependent manner. To account for this effect, we established calibration curves and corrected all fluorescence measures accordingly (as described in S9 Fig).

### Antibiotic-antivirulence combination assays

From the single drug dose-response curves, we chose for each drug nine concentrations (including no drugs) to cover the entire activity range in each medium, including no, intermediate, and strong inhibitory growth effects on PAO1 (S2 Table). We then combined these drug concentrations in a 9×9 matrix for each of the eight antibiotic-antivirulence pairs and repeated the growth experiment for all combinations in 6-fold replication, exactly as described above. After 48 hours of incubation, we measured growth and virulence factor production following the protocols described above.

### Synergy degree of drug combinations

We used the Bliss independence model to calculate the degree of synergy (*S*) for both growth and virulence factor inhibition, for each of the antibiotic-antivirulence combinations [84–86].

The Bliss model has been suggested as the best model of choice for drugs with different modes of actions and targets [85], as it is the case for antibiotics and antivirulence compounds. We used the formula $S = f_{x,0} \cdot f_{0,y} - f_{xy}$, where $f_{X,0}$ is the growth (or virulence factor production) level measured under antibiotic exposure at concentration X; $f_{0,Y}$ is the growth (or virulence factor production) level measured under antivirulence exposure at concentration Y; and $f_{X,Y}$ is the growth (or virulence factor production) level measured under the combinatorial treatment at concentrations $X$ and $Y$. If $S = 0$, then the two drugs act independently. Conversely, $S < 0$ indicates antagonistic drug interactions, while $S > 0$ indicates synergy.

## Experimental evolution under antibiotic treatment

To select for AtbR clones, we exposed overnight cultures of PAO1 WT (initial $OD_{600} = 10^{-4}$) to each of the four antibiotics in LB medium (antibiotic concentrations, ciprofloxacin: 0.15 μg/mL; colistin: 0.5 μg/mL; meropenem: 0.8 μg/mL; tobramycin: 1 μg/mL) in 6-fold replication. These antibiotic concentrations initially caused a 70%–90% reduction in PAO1 growth compared to untreated cultures, conditions that imposed strong selection for the evolution of resistance. The evolution experiment ran for seven days, whereby we diluted bacterial cultures and transferred them to fresh medium with the respective treatment, with a dilution factor of $10^{-4}$, every 24 hours. At the end of each growth cycle, we measured growth ($OD_{600}$) of the evolving lineages using a SpectraMax Plus 384 plate reader (Molecular Devices, Biberach, Germany).

## Phenotypic and genetic characterization of resistance

Following experimental evolution, we screened the evolved lines for the presence of AtbR clones. For each antibiotic, we plated four evolved lines on LB plates and isolated single clones, which we then exposed in liquid culture to the antibiotic concentration they experienced during experimental evolution. Among those that showed growth restoration (compared to the untreated WT), we picked two random clones originating from different lineages per antibiotic for further analysis. We had to adjust our sampling design in two cases. First, only one population survived our ciprofloxacin treatment and thus only one resistant clone could be picked for this antibiotic. Second, clones evolved under colistin treatment grew very poorly in CAS medium, and therefore we included an experimentally evolved colistin resistant clone from a previous study, which did not show compromised growth in CAS (see [38] for a description on the experimental evolution). Altogether, we had seven clones (one clone per antibiotic was allocated to one of the two media, except for ciprofloxacin). For all these clones, we re-established the drug-response curves in either CAA+Tf or CAS and quantified the $IC_{50}$ values (S4 Fig). For all cases, the $IC_{50}$ of the AtbR clones was significantly higher than that of the ones of the antibiotic sensitive WT ($-187.30 \leq t \leq -3.10$, $p < 0.01$ for all AtbR clones; S4 Fig). Furthermore, we examined whether resistance to antibiotics can lead to collateral sensitivity or cross-resistance to antivirulence compounds, and that is why we also established the dose-response curves for all the seven AtbR clones for the respective antivirulence compounds. As before, we exposed the clones and the WT PAO1 to a range of gallium (0–50 μM) or furanone (0–390 μM) concentrations in CAA+Tf or CAS, respectively, and compared their $IC_{50}$ values.

We further isolated the genomic DNA of the selected evolved AtbR clones and sequenced their genomes. We used the GenElute Bacterial Genomic DNA kit (Sigma-Aldrich, Buchs SG, Switzerland) for DNA isolation. DNA concentrations were assayed using the Quantifluor dsDNA sample kit (Promega, Dübendorf, Switzerland). Samples were sent to the Functional Genomics Center Zurich for library preparation (TruSeq DNA Nano) and sequencing on the Illumina MiSeq platform with v2 reagents and pair-end 150-bp reads. In a first step, we

mapped the sequences of our ancestral WT PAO1 strain (European Nucleotide Archive [ENA] accession number: ERS1983671) to the *P. aeruginosa* PAO1 reference genome (NCBI accession number: NC_002516) with snippy (https://github.com/tseemann/snippy) to obtain a list with variants that were already present at the beginning of the experiment. Next, we quality filtered the reads of the evolved clones with trimmomatic [87], mapped them to the reference genome, and called variants using snippy. Detected variants were quality filtered, and variants present in the ancestor strain were excluded from the dataset using vcftools [88]. The mapping files generated in this study are deposited in the ENA under the study accession number PRJEB32766.

## Monoculture and competition experiments between sensitive and resistant clones

To examine the effects of combination treatments on the growth and the relative fitness of AtbR clones, we subjected the sensitive WT PAO1 (tagged with GFP) and the experimentally evolved AtbR clones (Table 1), alone or in competition, to six different conditions: (i) no drug treatment; (ii) antibiotics alone; (iii–iv) two concentrations of antivirulence compounds alone; and (v–vi) the same two concentrations of antivirulence compounds combined with antibiotics. Antibiotic concentrations are listed in S2 Table, while antivirulence concentrations were as follows: gallium, 1.56 μM (low), 6.25 μM (intermediate); and furanone, 6.3 μM (low), 22.8 μM (intermediate). Bacterial overnight cultures were prepared and diluted as described above. Competitions were initiated with a mixture of 90% sensitive WT cells and 10% resistant clones to mimic a situation where resistance is still relatively rare. Mixes alongside monocultures of all strains were inoculated in either 200 μL of CAA+Tf or CAS under all the six treatment regimes. We used flow cytometry to assess strain frequency prior to and after a 24-hour competition period at 37˚C static (S10 Fig). Specifically, bacterial cultures were diluted in 1× phosphate buffer saline (PBS; Gibco, ThermoFisher, Zurich, Switzerland) and frequencies were measured with a LSRII Fortessa cell analyzer (BD Biosciences, Allschwil, Switzerland; GFP channel, laser: 488 nm, mirror: 505LP, filter: 530/30; side and forward scatter: 200-V threshold; events recorded with CS&T settings) at the Cytometry Facility of the University of Zurich. We recorded 50,000 events before competitions and used a high-throughput sampler device (BD Bioscience) to record all events in a 5-μL volume after competition. Because antibacterials can kill and thereby quench the GFP signal in tagged cells, we quantified dead cells using the propidium iodide (PI) stain (2 μL of 0.5 mg/mL solution) with flow cytometry (for PI fluorescence: laser: 561 nm, mirror: 600LP, filter: 610/20).

We used the software FlowJo (BD Biosciences, Ashland, OR) to analyze data from flow cytometry experiments. We followed a three-step gating strategy: (i) we separated bacterial cells from media and noise background by using forward- and side-scatter values as a proxy for particle size; (ii) within this gate, we then distinguished live from dead cells based on the PI staining; (iii) finally, we separated live cells into GFP-positive and -negative populations. Fluorescence thresholds were set using appropriate control samples: isopropanol-killed cells for PI-positive staining and untagged PAO1 cells for GFP-negative fluorescence. We then calculated the relative fitness of the AtbR clone as $\ln(v) = \ln\{[a_{24}\times(1-a_0)]/[a_0\times(1-a_{24})]\}$, where $a_0$ and $a_{24}$ are the frequencies of the resistant clone at the beginning and at the end of the competition, respectively [89]. Values of $\ln(v) < 0$ or $\ln(v) > 0$ indicate whether the frequency of AtbR clones decreased or increased relative to the sensitive PAO1-GFP strain. To check for fitness effects caused by the fluorescent tag, we included a control competition, where we mixed PAO1-GFP with the untagged PAO1 in a 9:1 ratio for all treatment conditions. We noted that high drug concentrations significantly curbed bacterial growth, which reduced the number of

events that could be measured with flow cytometry. This growth reduction increased noise relative to the signal, leading to an overestimation of the GFP-negative population in the mix. To correct for this artifact, we established calibration curves for each individual experimental replicate for how the relative fitness of untagged PAO1 varies as a function of cell density in control competitions with PAO1-GFP. Coefficients of the asymptotic functions used for the correction are available together with the raw dataset.

## Statistical analysis

All statistical analyses were performed with RStudio v.3.3.0 [90]. We fitted individual dose-response curves with either log-logistic or Weibull functions, and estimated and compared $IC_{50}$ values using the drc package [91], while dose-response curves under the combination treatment were fitted using spline functions. We used Pearson correlation coefficients to test for significant associations between the degree of synergy in growth and virulence factor inhibition. We used one-sample *t* tests to compare the degree of synergy of each combination of concentrations represented in Fig 4 to zero. We used Welch's two-sample *t* test to compare growth between the sensitive WT PAO1 and the resistant clones under antibiotic treatment. To compare the relative fitness of resistant clones to the reference zero line, we used one-sample *t* tests. Finally, we used ANOVA to test whether the addition of antivirulence compounds to antibiotics affected the relative fitness of AtbR clones, and whether the outcome of the competition experiment is associated with the degree of synergy of the drug combinations. Where necessary, *p*-values were adjusted for multiple comparisons using the false discovery rate method.

## Supporting information

**S1 Fig. Effect of antibiotics on virulence factor production in *P. aeruginosa* PAO1 populations.** We exposed PAO1 to all four antibiotics in two the experimental media: CAA+Tf (iron-limited CAA with transferrin) and CAS. After 48 hours of exposure, we measured virulence factor production: pyoverdine in CAA+Tf and proteases in CAS. The inhibition of virulence factors followed the same pattern as for growth inhibition, except for ciprofloxacin and meropenem, where pyoverdine production slightly increased at intermediate antibiotic concentrations and only dropped at higher antibiotic levels. Dots show means ± standard error across six replicates. All data are scaled relative to the drug-free treatment. Data stem from the same two independent experiments as shown in Fig 1. The red dots indicate the highest concentration used for each experiment, from which 7 serial dilution steps were tested. Curves were fitted with log-logistic functions. The underlying data for this figure can be found at https://doi.org/10.6084/m9.figshare.12515364. CAA, casamino acid medium; CAS, casein medium; Tf, human apo-transferrin.
(TIF)

**S2 Fig. Statistical significance maps for antibiotic-antivirulence drug interactions.** For each drug concentration combination, we tested whether the degree of synergy is significantly different from zero (i.e., independent drug interaction). Heatmaps depict *p*-values ranging from white (no significant drug interaction) to blue (significant antagonism) to red (significant synergy). *p*-Values are shown for gallium-antibiotic combinations (**A-D** for growth; **E-H** for pyoverdine production) and furanone-antibiotic combinations (**I-L** for growth; **M-P** for protease production). To account for multiple comparisons, we corrected the *p*-values for each drug combination using the "false discovery rate" method. The underlying data for this figure can be found at https://doi.org/10.6084/m9.figshare.12515364.
(TIF)

**S3 Fig. Assessing the relationship between the degrees of synergy for growth and virulence factor inhibition.** We found that the degrees of synergy for the two measured traits across the 9×9 antibiotic-antivirulence combination matrix correlated in 6 out of 8 cases (Pearson correlation coefficient: ciprofloxacin-gallium: $r = 0.09$, $t_{79} = 0.85$, $p = 0.394$; colistin-gallium: $r = 0.69$, $t_{79} = 8.51$, $p < 0.001$; meropenem-gallium: $r = 0.17$, $t_{79} = 1.52$, $p = 0.130$; tobramycin-gallium: $r = 0.58$, $t_{79} = 6.39$, $p < 0.001$; ciprofloxacin-furanone: $r = 0.34$, $t_{79} = 3.22$, $p = 0.002$; colistin-furanone: $r = 0.96$, $t_{79} = 32.50$, $p < 0.001$; meropenem-furanone: $r = 0.87$, $t_{79} = 15.48$, $p < 0.001$; tobramycin-furanone: $r = 0.75$, $t_{79} = 10.16$, $p < 0.001$). Solid lines show association trend lines between the two levels of interactions. The underlying data for this figure can be found at https://doi.org/10.6084/m9.figshare.12515364.
(TIF)

**S4 Fig. Clones evolved under antibiotic treatments show altered dose-response curves.** To confirm that the evolved clones are resistant to the antibiotic in the two experimental media (iron-limited [CAA+Tf] and CAS media), we measured for each antibiotic the dose-response curves for the ancestral WT and one randomly selected clone either in CAA+Tf or CAS. For each antibiotic and medium, we tested 11 concentrations within these ranges: ciprofloxacin: 0–1 µg/mL (CAA+Tf), 0–4 µg/mL (CAS); colistin: 0–1.2 µg/mL (CAA+Tf), 0–3.33 µg/mL (CAS); meropenem: 0–1.6 µg/mL (CAA+Tf), 0–7 µg/mL (CAS); tobramycin: 0–2.4 µg/mL (CAA+Tf), 0–24 µg/mL (CAS). All evolved clones showed an attenuated dose-response curve, were able to grow at higher drug concentrations than the ancestral WT and had significantly higher $IC_{50}$ values. Measurements of $OD_{600}$ were taken after 48 hours incubation time at 37°C under static conditions. Growth values are scaled relative to the untreated control for each strain. Data are shown as means ± standard errors across four replicates. Curves were fitted with four parameters log-logistic functions. In the lower left corner of each panel we show the mean $IC_{50}$ values of the WT and AtbR clones ± standard errors (µg/mL) and the respective statistical analysis comparing the ratio of the means. The underlying data for this figure can be found at https://doi.org/10.6084/m9.figshare.12515364. AtbR clones, antibiotic resistant clones; CAA, casamino acid medium; CAS, casein medium; $IC_{50}$, half maximal inhibitory concentration; $OD_{600}$, optical density at 600 nm; Tf, human apo-transferrin; WT, wild-type.
(TIF)

**S5 Fig. Antivirulence dose-response curves for AtbR clones of *P. aeruginosa* PAO1.** To check whether resistance to antibiotics influenced the susceptibility to antivirulence compounds, we exposed our selected AtbR clones to a range of concentrations of both gallium (0–50 µM) and furanone (0–390 µM). Under gallium treatment, only the colistin resistant clone showed increased sensitivity to the antivirulence drug. Under furanone treatment, the clones resistant to ciprofloxacin and colistin showed a certain level of cross-resistance to this antivirulence compound, while we found collateral sensitivity between tobramycin and furanone. All values are scaled relative to the untreated control for each strain, and data points show the mean across four replicates. We used either log-logistic functions (in CAA+Tf) or three-parameter Weibull functions (in CAS) to fit the curves and extract mean $IC_{50}$ values ± standard error, which are reported in the left bottom corner of each panel together with the respective statistical analysis comparing the ratio of the means. The underlying data for this figure can be found at https://doi.org/10.6084/m9.figshare.12515364. AtbR clones, antibiotic resistant clones; CAA, casamino acid medium; CAS, casein medium; $IC_{50}$, half maximal inhibitory concentration; $OD_{600}$, optical density at 600 nm; Tf, human apo-transferrin; WT, wild-type.
(TIF)

**S6 Fig. Testing for correlations between the degree of synergy for growth inhibition and the outcome of competitions.** We tested whether the degree of synergy for growth inhibition is a predictor of the competition outcome between the AtbR clones and the susceptible WT under combination treatment. We compared the degrees of synergy of each drug combination for the AtbR clones (**A**) or the WT (**B**) to their relative fitness values in competition. Positive or negative y-values indicate that the clones increased or decreased in frequency during the competition, respectively. Positive or negative values on the x-axis indicate synergy or antagonism, respectively. There were no significant associations between the relative fitness and the degree of synergy for growth inhibition neither for the AtbR clones nor for the WT (ANOVA, for AtbR: $F_{1,65} = 0.88$, $p = 0.353$; for WT: $F_{1,65} = 1.85$, $p = 0.179$). Instead, relative fitness was significantly affected by the type of antivirulence drug (ANOVA, for AtbR clones: $F_{1,65} = 106.36$, $p < 0.001$; for WT: $F_{1,65} = 44.58$, $p < 0.001$) and the specific antibiotic-antivirulence combination applied (ANOVA, for AtbR clones: $F_{3,65} = 37.45$, $p < 0.001$; for WT: $F_{3,65} = 14.50$, $p < 0.001$). The underlying data for this figure can be found at https://doi.org/10.6084/m9.figshare.12515364. AtbR clones, antibiotic resistant clones; WT, wild-type.
(TIF)

**S7 Fig. Validation of mCherry fluorescence as a proxy for growth measurements.** The CAS media has a very high turbidity due to the poor solubility of casein, which interferes with $OD_{600}$, which is typically used as a measure of growth. We therefore used mCherry fluorescence, constitutively expressed from a single-copy chromosomal insertion, as a proxy for bacterial growth in CAS. To validate this method, we grew PAO1-mCherry (able to digest CAS) and PAO1 *ΔlasR*-mCherry (unable to digest CAS) in CAS medium for 48 hours at 37°C in a Tecan plate reader tracking $OD_{600}$ and mCherry fluorescence every 15 minutes. (**A**) Blank corrected $OD_{600}$ trajectories for PAO1-mCherry and PAO1 *ΔlasR*-mCherry. PAO1 *ΔlasR*-mCherry grew poorly but showed a standard sigmoid growth pattern by digesting the supplemented CAA. In stark contrast, the $OD_{600}$ of PAO1-mCherry first increased sharply, then declined dramatically, followed by a slow linear increase over time. This trajectory is explained by the simultaneous growth of bacteria (increasing $OD_{600}$) and clearance of the turbidity due to protein digestion (decreasing $OD_{600}$), thus demonstrating that $OD_{600}$ is an unsuitable measure for growth. (**B**) Blank corrected mCherry trajectories for PAO1-mCherry and PAO1 *ΔlasR*-mCherry. As for $OD_{600}$, PAO1 *ΔlasR*-mCherry grew only poorly (according to the mCherry signal) and only within the first 7 hours of the assay, digesting the supplemented CAA. Unlike for $OD_{600}$, the mCherry signal yielded a much more sensible growth trajectory for PAO1-mCherry, characterized by an initial increase (CAA consumption), followed by a lag phase (protease secretion and switch to CAS) and growth resumption (CAS digestion). (**C**) To further validate that mCherry fluorescence is a good proxy for growth in CAS media, we compared the endpoint measurements of mCherry fluorescence with CFU/mL values, determined by plating the cultures on LB-agar plates. All values are scaled relative to PAO1-mCherry. The two methods yielded similar results and show that the growth of PAO1*ΔlasR*-mCherry is approximately 25% of the one of PAO1-mCherry. Data are shown as means ± standard errors across eight replicates for the growth curves and eight (PAO1-mCherry) or four (PAO1*ΔlasR*-mCherry) replicates for the CFU/mL data. The underlying data for this figure can be found at https://doi.org/10.6084/m9.figshare.12515364. CAA, casamino acid medium; CAS, casein medium; CFU, colony forming units; LB, Lysogeny broth; $OD_{600}$, optical density at 600 nm.
(TIF)

**S8 Fig. Correlation between endpoint measurements and area under the growth curve (integral) for single drug treatments.** To verify that a single $OD_{600}$ or mCherry measurement after growth is a good proxy for growth inhibition, we tested the correlation between the area

under the growth curve (integral) and endpoint measurements, under single drug treatments in CAA+Tf (**A**) or CAS (**B**) media. For each antibiotic and antivirulence compound, we picked 9 concentrations that cover the entire drug active range and that were used for the combination assay, shown in Figs 3 and 4. Each concentration was tested in 5-fold replication. Cultures were grown for 48 hours in a Tecan Infinite M-200 plate reader (Tecan Group, Switzerland) and growth was recorded by reading $OD_{600}$ (in CAA+Tf) or mCherry fluorescence (in CAS) every 15 minutes, after a short shaking event. Growth trajectories were established with a spline fit, and the two parameters (endpoint yield and integral) were extracted using the grofit package in RStudio. In both media, the two growth parameters showed strong linear association patterns (blue lines and $R^2$ values). For several drugs, growth integral measurements were more sensitive to discover growth inhibitions at low drug concentrations (light gray circles), and that is why cubic data fits (red lines and $R^2$ values) often explained an even higher proportion of the variance. Nonetheless, these control analyses show that endpoint growth values are reliable proxies for measuring growth inhibition under drug treatment. The underlying data for this figure can be found at https://doi.org/10.6084/m9.figshare.12515364. CAA, casamino acid medium; CAS, casein medium; $OD_{600}$, optical density at 600 nm; Tf, human apo-transferrin.
(TIF)

**S9 Fig. Fluorescence correction for mCherry and pyoverdine measurements in the presence of furanone C-30 and gallium.** The two metals bromine (in furanone C-30) and gallium interfere with the fluorescence measurements of mCherry and pyoverdine in a concentration-dependent manner. To account for this bias, we established calibration curves and used them to correct fluorescent values in all experiments. (**A**) Furanone C-30 is autofluorescent in the mCherry channel (excitation 582 nm, emission 620 nm). We quantified the autofluorescence in function of the concentration of furanone both in CAS and CAA media. Briefly, we incubated each media supplemented with a range of furanone C-30 concentrations (0–390 μM, as used in Fig 2, in 6-fold replication) for 48 hours under static conditions and then measured mCherry fluorescence. The relationship between concentration and fluorescence was explained by a four-parameter logistic function in CAS or by a three-parameter Gompertz function in CAA. In all experiments, we used this calibration curve to subtract, for each furanone concentration, the autofluorescence component from the mCherry measurements. (**B**) The fluorescent signal of pyoverdine becomes inflated when gallium binds to the siderophore [27,38]. We used the supplementary data from Ross-Gillespie and colleagues [27] to quantify this bias in fluorescence as a function of gallium concentration. They incubated 200 μM pyoverdine in iron-limited CAA+Tf medium, supplemented with gallium concentrations ranging from 0 to 1 mM, and measured pyoverdine-associated fluorescence. When supplemented with more than 50 μM gallium, pyoverdine showed a nearly 2-fold higher fluorescence signal. This signal bias can be explained by a five-parameter log-logistic function. In all our experiments, for each gallium concentration used, we applied correction factors derived from this fitted curve to account for this potential bias. The underlying data for this figure can be found at https://doi.org/10.6084/m9.figshare.12515364. CAA, casamino acid medium; CAS, casein medium; Tf, human apo-transferrin.
(TIF)

**S10 Fig. Examples of flow cytometry scatterplots from competition experiments between the sensitive WT PAO1 and AtbR clones.** The WT strain PAO1, chromosomally tagged with a constitutively expressed GFP marker, was co-cultured with AtbR clones in a 90:10 ratio in the presence of five different drug treatments. Mono- and mixed cultures were measured with the flow cytometer at the beginning (time, 0 hours) and at the end (time, 24 hours) of the

competition experiments. For data analysis, we plotted the size of the cells (forward scatter, FSC) against the GFP fluorescence to distinguish tagged from untagged cells. The shown plots depict an illustrative example, in which ciprofloxacin was used as an antibiotic. (**A**) A mono-culture of the untagged AtbR clone does not show GFP fluorescence. This control allows quantifying the background fluorescence of the cells. (**B**) A monoculture of the tagged PAO1-GFP shows relatively strong GFP fluorescence, with 99.1% of all cells considered as GFP positive. (**C**) In a 90:10 volumetric mix of WT and AtbR clones, the cells of the two strains can be unambiguously distinguished and their actual ratio (88:12) can be determined. Frequencies of GFP-positive and GFP-negative cells were then quantified after a 24-hour incubation period at 37°C. (**D**) The monoculture of the untagged AtbR strain shows that cells do not increase their GFP autofluorescence over time, and 100% of cells fall into the GFP-negative gate. (**E**) The monoculture of PAO1-GFP shows relatively strong fluorescence also at the end of the competition, with 99.3% of cells being classified as GFP positive. (**F**) The mix of WT and AtbR clones, when grown in absence of any drug treatment stays at the initial frequency (88.2:11.8). (**G**) When the mix was grown in the presence of the antibiotic, the fraction of untagged AtbR strain increases to 23.6%, demonstrating their selective advantage. AtbR clones, antibiotic resistant clones; GFP, green fluorescent proteins; WT, wild-type.
(TIF)

**S1 Table. Statistical analyses (*t* test and ANOVA) performed on the data presented in Fig 6.**
(DOCX)

**S2 Table. Concentrations of antibiotics and antivirulence drugs used for the combinatorial treatments and the competition assays.**
(DOCX)

## Acknowledgments

We thank Roland Regös and Désirée Bäder for advice on the Bliss model; Alex Hall, Roland Regös, and Frank Schreiber for comments on the manuscripts; David Wilson for experimental support; Selina Niggli, Priyanikha Jayakumar, and the Flow Cytometry Facility (University of Zurich) for support with flow cytometry experiments; and the Functional Genomics Center Zurich for technical support with the strain sequencing.

## Author Contributions

**Conceptualization:** Chiara Rezzoagli, Martina Archetti, Rolf Kümmerli.

**Data curation:** Chiara Rezzoagli, Martina Archetti, Ingrid Mignot.

**Formal analysis:** Chiara Rezzoagli, Martina Archetti, Ingrid Mignot, Michael Baumgartner, Rolf Kümmerli.

**Funding acquisition:** Rolf Kümmerli.

**Methodology:** Chiara Rezzoagli, Martina Archetti, Ingrid Mignot, Michael Baumgartner.

**Supervision:** Rolf Kümmerli.

**Validation:** Chiara Rezzoagli, Martina Archetti, Ingrid Mignot, Rolf Kümmerli.

**Writing – original draft:** Chiara Rezzoagli, Martina Archetti, Michael Baumgartner, Rolf Kümmerli.

**Writing – review & editing:** Chiara Rezzoagli, Martina Archetti, Ingrid Mignot, Michael Baumgartner, Rolf Kümmerli.

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
