## [Editor Report · Decision Letter 0]

9 Dec 2019

Dear Dr Kuemmerli, 

Thank you for submitting your manuscript entitled "Combining antibiotics with antivirulence compounds is effective and can reverse selection for antibiotic resistance in Pseudomonas aeruginosa" for consideration as a Research Article by PLOS Biology.

Your manuscript has now been evaluated by the PLOS Biology editorial staff as well as by an academic editor with relevant expertise and I am writing to let you know that we would like to send your submission out for external peer review.

Please re-submit your manuscript within two working days, i.e. by Dec 11 2019 11:59PM.

Kind regards,

Lauren A Richardson, Ph.D

Senior Editor

PLOS Biology

---

## [Decision Letter · Decision Letter 1]

2 Jan 2020

Dear Dr Kuemmerli,

Thank you very much for submitting your manuscript "Combining antibiotics with  antivirulence compounds is effective and can reverse selection for antibiotic resistance in Pseudomonas aeruginosa" for consideration as a Research Article at PLOS Biology. Your manuscript has been evaluated by the PLOS Biology editors, an Academic Editor with relevant expertise, and by several independent reviewers.

In light of the reviews (below), we will not be able to accept the current version of the manuscript, but we would welcome re-submission of a much-revised version that takes into account the reviewers' comments. The Academic Editor has provided comments, included below the reviews, to help guide your revision. We cannot make any decision about publication until we have seen the revised manuscript and your response to the reviewers' comments. Your revised manuscript is also likely to be sent for further evaluation by the reviewers.

We expect to receive your revised manuscript within 2 months. 

**IMPORTANT - SUBMITTING YOUR REVISION**

*NOTE: In your point by point response to to the reviewers, please provide the full context of each review. Do not selectively quote paragraphs or sentences to reply to. The entire set of reviewer comments should be present in full and each specific point should be responded to individually, point by point.

*Re-submission Checklist*

*Published Peer Review*

*PLOS Data Policy*

*Blot and Gel Data Policy*

Sincerely,

Lauren A Richardson, Ph.D

Senior Editor

PLOS Biology

REVIEWS:

Reviewer #1: 

The aim of this study is to systematically assess the potential of novel combination treatments in which antibiotics (AB) are used together with antivirulence compounds (AV). AB+AV treatments are highly interesting as they may help to combat antibiotic resistance. 

I enjoyed reading this piece of research. The experiments are straightforward and the manuscript is well written (clear and accessible to a wide audience). The authors focus on two AVs with distinct mechanisms (furanone, gallium) which are combined with four clinically relevant AB (CIP, MER, COL, TOB). 

The authors claims are that 

1) Certain AB+AV combinations produce clinically relevant levels of synergy

2) The interaction type (synergy, independent, antagonistic) is concentration dependent

3) AV can re-sensitize AB-resistant bacteria

4) AV+AB can select against resistance

These findings would be a significant contribution and suitable for publication with PLOS Biology. However, claims 3 + 4 are not sufficiently supported by the data, as important controls were missing (please see major point 1 below). I also have some technical concerns regarding the AV quantification (major point 2). 

Major points

MP1. It is unclear whether the observed effects of re-sensitization and negative selection (Fig 5) are caused by drug interaction (synergy), or alternatively, and currently not considered by the authors, by the inhibitory effect of the AVs. This question can be addressed by experiments a, b. 

a) Measure growth for treatments with AV alone (no antibiotic added) at the concentrations used in Fig 5A.

b) Competition experiments with AV alone (no antibiotics added) as an addition to Fig 5B.

Antibiotic resistant strains frequently show hypersensitivity to other antibiotics (so-called collateral sensitivity). It is possible that similar phenomenon exists for AV, which may explain for example the negative selection of TOB-resistant bacteria in furanone-TOB combinations. This can be addressed by experiment c.

c) AV dose-response analyses similar to Fig 2 for the resistant strains.

MP2. My understanding of virulence quenching is that the bacteria produce less AV per cell. However, it seems to me that the authors report the total amounts of virulence compound. As evident from Fig 2 there is a strong positive correlation between total AV amount and cell density. It is therefore unclear whether the antivirulence quenching worked in the combinations (Figs 3 and 4). A possible solution would be to normalize the relative AV amount by relative growth. 

MP3. It is unclear whether cell density could affect the quantification of pyoverdine.

The paper provides enough details of its methodology so that its experiments could be reproduced. 

The authors treat the previous literature fairly.

Minor points: 

An explanation of why the observed effects are interpreted to provide "clinically interesting levels of synergy" (line 29) would be helpful.

The words "is effective" in the title are unclear.

Line 318: for clarity please replace "induce" with "confer"

Please rationalize, why Bliss independence was used compared to Loewe additivity.

-------------

Reviewer #2: 

To address the burning issue of antibiotic crisis, combination therapies are more and more recommended. Often, combination therapy is thought of as a combination of two antibiotics and the, mostly untested, rationale is that it is more difficult for bacteria two evolve two resistance mechanisms simultaneously. Here, the authors take a different approach. They investigate the efficacy of and the resistance evolution against a combination between antibiotics and adjuvants. Theses adjuvants are based on anti-virulence compounds. While, as the authors point out, such anti-virulence compounds have been investigated before also in combinations with antibiotics, the current manuscript clearly goes beyond that. It systematically, across concentration gradients, evaluates interactions two-way interactions between two anti-virulence compounds in three relevant antibiotics and one relevant antimicrobial peptides (colistin). Importantly, this is combined with experimental evolution. The re-sequencing of the resultant strains also reveals that the mutations underlying the evolved resistant are those found from clinical strains. 

Overall, this is a well-crafted manuscript with very interesting results. I feel that the manuscript would benefit from some clarifications and brief additional information in places:

Line 94 Please explain why you analyze symbiosis with Bliss independence and not for example with a Loewe approach. I have no issue with Bliss, it would just be good to know. 

Line 101: Here, cross resistance is mentioned. But I cannot find data on cross-resistance. The authors infer the potential of cross-resistance from the mutations that result under experimental selection. This is fine, and I don't want to argue for additional experiments, but would suggest toning down cross-resistance here. Otherwise it would be useful to present data on cross-resistance of the selected lines. 

Line 176. There is not much detail given on the selection lines. It would be nice if the authors provided basic pharmacodynamic data in the supplement, such as MIC and perhaps also lag phase and growth rates (I assume they have got those data from their plate reader essays).

Line 224-250. It would be nice to provide a bit more information here. For example, on line 227 a singular is used and then it looks like more mutants are studied. Also, for Colistin 2 clones are mentioned, but more parallel lines are selected. Please make clear what the sample sizes are. Also, here background information is provided on the resistance mutations. While these resistance mutations fall within resistance mutations that have been described, they obviously do not cover the full spectrum that is known. Therefore it would be desirable in the discussion, when the authors hone in on particular combinations to make clear that this is also related to particular mutations and perhaps make a prediction on what would happen if other common mutations were involved. 

Line 269. I can see that the combinations tested here are potentially interesting. But has the toxicity of the anti-virulence compounds for patients been investigated. Could be briefly mentioned. 

Lines 277-291. The point that the nature of interactions is concentration dependent is important. Perhaps it would be worth mentioning pharmacokinetics of drugs within patients which will determine the nature of interactions within patients. 

Lines 311-312. I have already mentioned this above. The resistance mechanisms of the experimentally evolved strains make cross-resistance likely in some cases. It would have been desirable to test this, but as the paper is rich in data, I would suggest to at least discuss the issue of cross-resistance in more detail. 

Fig. S1: I find it unusual to present correlations with a fitted regression line.

-------------

Reviewer #3: 

The authors assess growth inhibition and antibiotic resistance selection under multiple combinations of antibiotic and anti-virulence drugs, and report a fascinating mix of results. The most promising of these results are truly eye-catching: synergy (ie less total drug required for same effect) *AND* re-sensitization of AMR strains to antibiotics (ie AVs can be anti-resistance adjuvants) *AND* selection against antibiotic resistance. These are exciting claims and therefore require a high bar for acceptance in my view. Below are some of my concerns that I would want to see addressed - 

Address clinical relevance sooner and more seriously in the MS (this is the main issue).

At present, theres a default sentence or two at the end on 'more work needed' towards clinical relevance. Given the motivation for this work is clinical, there needs to be better integration and earlier on. Let's take the best case, tobramycin + AVs, here the authors describe the striking combination of synergy, antibiotic repotentiation and reversal of resistance selection (WOW!). But the authors also report 'interaction hot-spots dependent on dose'. So the simple and addressable clinical relevance question becomes - do these exciting interactions happen under clinically relevant in vivo concentrations? This would require adding some in vivo benchmarks on the antibiotic doses (e.g. in vivo max; in vivo mean). These are upper thresholds - but it is also possible that these striking effects happen under relatively low dosing that (even with a synergy boost) are below reasonable clinical dosing goals. Both directions of clinical irrelevance should be directly assessed. I would also encourage the authors to consider taking their most promising results into a mouse model. They mention that in vivo would introduce spatial structure. That is just the start of it - in vivo would allow the virulence factors to actually have a virulence phenotype, plus impacts of immunity, pharmacodynamics, relevant nutritional environment, etc etc. 

Summarize a priori hypotheses at the front of the MS. 

At present the MS is a bit of a laundry-list in places. Basically, the authors test a ton of interactions without stating clear testable hypotheses up front. This would help to sort the reams of data into results that are consistent with our current expectations and ones that are surprises (and therefore require careful attention). 

Resistance to AV? 

Another path to treatment failure would be resistance to the AV component of a combination therapy. This is not assessed in this MS. 

Synergy significance threshold. 

Lines 120-160 ish. Lots of statements on independence / non-independence, but no statistical support. What was the statistical test to support the claim that x-y was independent or synergistic under a given dosing, etc? multiple testing has to be considered. 

Re-sensitization

There's a sentence referencing the re-sensitization figure (fig 5A) that states that the AV doses have little direct effect on growth. This should be made clearer and quantified in the figure or figure legend. I attempted to gauge what direct effect the 'high' AV dose in fig 5A would have by referencing fig 2, and hard to get a clear figure due to log scale but could be 50% growth reduction which is substantial. The authors should spell out the direct effect here.

-------------

COMMENTS FROM THE ACADEMIC EDITOR

I agree with the generally positive evaluation of the manuscript by the three reviewers. Thus I concur that the submitted manuscript is in principle suitable for PLoS Biol. I also agree that the manuscript should be further improved before it can be accepted. In particular, the suggestions from reviewer #1 for further experiments and data analyses should be considered (major points 1-3 from reviewer #1), as they would help to clarify some of the open points in the results. All three reviewers made helpful suggestions and comments as to how the text could be improved. These should be taken into account while revising the manuscript. I also agree with reviewer #3 that the clinical relevance of the data is not convincing. However, additional mouse experiments or clinical studies would go beyond the scope of the current study and the new insights obtained are already of sufficient importance as they may point to a new treatment strategy. Nevertheless, the authors should reconsider how they link their study to clinical reality and avoid any unrealistic claims that the current data does not support. The additional suggestions from reviewer #3 for further experiments and data analyses (see last three comments from reviewer #3) are very helpful and should be considered.

---

## [Decision Letter · Decision Letter 2]

9 Jun 2020

Dear Dr Kuemmerli,

Thank you for submitting your revised Research Article entitled "Combining antibiotics with antivirulence compounds can have synergistic effects and reverse selection for antibiotic resistance in Pseudomonas aeruginosa" for publication in PLOS Biology. I have now obtained advice from the original reviewers and have discussed their comments with the Academic Editor. 

As you can see, both reviewers feel the manuscript is significantly improved and have only minor comments. Rev. 3 does note however that his original request for clinical dose benchmarks has not been addressed. Based on the reviews, we will probably accept this manuscript for publication, assuming that you will modify the manuscript to address the remaining points raised by the reviewers. Please make sure to address the minor edits from Rev. 1 but also directly and explicitly discuss any limitations regarding clinical relevance. IMPORTANT: Please also make sure to address the data and other policy-related requests noted at the end of this email.

We expect to receive your revised manuscript within two weeks. Your revisions should address the specific points made by each reviewer. In addition to the remaining revisions and before we will be able to formally accept your manuscript and consider it "in press", we also need to ensure that your article conforms to our guidelines. A member of our team will be in touch shortly with a set of requests. As we can't proceed until these requirements are met, your swift response will help prevent delays to publication.

---

*Copyediting*

*Published Peer Review History*

*Early Version*

*Submitting Your Revision*

Sincerely,

Hashi Wijayatilake, PhD, 

Managing Editor

PLOS Biology

DATA POLICY:

Figs. 1, 2AB, 3AB, 4A-P, 5AB, 6, and all ten supplementary figures (Figs. S1-S10)

FURTHER REQUESTS:

- Please complete your Data Availability Statement and provide the Figshare accession number to ensure that your Data Statement in the submission system accurately describes where your data can be found.

REVIEWS:

Reviewer #1: 

I would like to congratulate the authors on a nice piece of research. I very much appreciate the additional experiments that were performed for the revisions. The new data now clarifies the previously open points, especially those regarding re-sensitization and selection inversion. 

Altogether, I highly recommend the publication of this research with PLOS Biology. 

I do not have further scientific criticisms. Some very minor points that the authors may wish to edit:

1) L. 91: I suggest to remove "antagonistically", as this could be confused with "antagonism"

2) L. 236: correct verb to "runs" or "ran"

3) Fig 1: X-axis label has "Concentration" in caps, in contrast to the other figures

---

Reviewer #3: 

The authors have responded well to most of my earlier concerns, and I'm overall positive about this MS. However there's one important point that I mentioned in my first review that has not been addressed: clinical dose benchmarks. 

I previously asked for clinical benchmarks, so that readers can begin to gauge whether the interesting phenomena are happening under conditions that are clinically relevant - ie between susceptible strain MICs and clinical break-points. This is a simple request and easy to address. For example for Tob, the question becomes, does the interesting behavior happen in the window of ~2 to 8 ug/ml? that's roughly the window between the MIC for wimpy PAO1 and a clinical breakpoint for resistance (sometimes higher). Unfortunately the PDF has been rendered by the journal with low resolution so I cant easily read the scale on fig 4 (not the authors fault), but looks like the synergy stops above zero-point-something ug/ml, ie below any relevant dose. If the interesting behavior for all drugs is outside of clinically relevant conditions, the paper is still interesting but the context and broader relevance does shift away from the clinic.

---

## [Editor Report · Decision Letter 3]

14 Jul 2020

Dear Dr Kuemmerli,

On behalf of my colleagues and the Academic Editor, Dr. Hinrich Schulenburg, I am pleased to inform you that we will be delighted to publish your Research Article in PLOS Biology. 

Early Version

PRESS 

Kind regards,

Pamela Berkman,

Publishing Editor

PLOS Biology

on behalf of

Roland Roberts,

Senior Editor

PLOS Biology